# *Escherichia coli* swimming is robust against variations in flagellar number

**Patrick J Mears[1,2], Santosh Koirala[3], Chris V Rao[3], Ido Golding[1,2,4]\*, Yann R Chemla[1,2]\***

[1]Department of Physics, University of Illinois at Urbana-Champaign, Urbana, United States; [2]Center for the Physics of Living Cells, University of Illinois at Urbana-Champaign, Urbana, United States; [3]Department of Chemical and Biomolecular Engineering, University of Illinois at Urbana-Champaign, Urbana, United States; [4]Verna and Mars McLean Department of Biochemistry and Molecular Biology, Baylor College of Medicine, Houston, United States

**Abstract** Bacterial chemotaxis is a paradigm for how environmental signals modulate cellular behavior. Although the network underlying this process has been studied extensively, we do not yet have an end-to-end understanding of chemotaxis. Specifically, how the rotational states of a cell's flagella cooperatively determine whether the cell 'runs' or 'tumbles' remains poorly characterized. Here, we measure the swimming behavior of individual *E. coli* cells while simultaneously detecting the rotational states of each flagellum. We find that a simple mathematical expression relates the cell's run/tumble bias to the number and average rotational state of its flagella. However, due to inter-flagellar correlations, an 'effective number' of flagella—smaller than the actual number—enters into this relation. Data from a chemotaxis mutant and stochastic modeling suggest that fluctuations of the regulator CheY-P are the source of flagellar correlations. A consequence of inter-flagellar correlations is that run/tumble behavior is only weakly dependent on number of flagella.

\*For correspondence: igolding@illinois.edu (IG); ychemla@illinois.edu (YRC)

## Introduction

Many species of bacteria swim by rotating helical filaments called flagella (*Berg, 2004*). A typical *Escherichia coli* cell is propelled by a bundle composed of multiple flagella. Each flagellum is controlled by a rotary motor that can switch between clockwise (CW) and counter-clockwise (CCW) rotation. When flagella on a cell rotate CCW, the cell swims along an approximately straight path called a 'run'. When some of the flagella rotate CW, the bundle is disrupted causing an abrupt change in direction called a 'tumble' (*Macnab and Ornston, 1977*). *E. coli* modulates the probability of being in one of these two swimming states in response to its environment, allowing it to navigate chemical, temperature, and light gradients (*Berg and Brown, 1972*; *Berg, 2004*). At any point in time, the probability that a flagellar motor rotates CW is determined by the concentration of phosphorylated signaling protein CheY (CheY-P). Coupling CheY phosphorylation to chemicals from the environment allows the cell to bias its random walk and migrate towards more favorable conditions. This biased random walk is called chemotaxis, and serves as a model for understanding how living organisms process information (*Berg and Brown, 1972*; *Wadhams and Armitage, 2004*; *Shimizu et al., 2010*).

Tremendous progress has been made towards elucidating the mechanism of bacterial chemotaxis. The relationship between the chemotaxis signaling network and the CCW/CW rotational bias of the individual flagellar motor is now well mapped ([*Block et al., 1982*; *Cluzel et al., 2000*; *Sourjik and Berg, 2002*; *Yuan et al., 2012*]; for a review see *Berg, 2004*), and has also been described using detailed mathematical models (*Emonet et al., 2005*; *Jiang et al., 2010*; *Shimizu et al., 2010*). Despite this wealth of knowledge, how the CCW/CW states of individual motors collectively determine the

**eLife digest** *Escherichia coli* is a rod-shaped bacterium commonly found in the lower intestines of humans and other warm-blooded animals. While most strains of *E. coli* are harmless, including most of those found in the human gut, some can cause diseases such as food poisoning. Due to its close association with humans and the fact that it is easy to grow and work with in the laboratory, *E. coli* has been intensively studied for over 60 years.

Many bacteria are capable of 'swimming' by using one or more flagella. These rotating whip-like structures are each driven by a reversible motor, and they act a bit like a propeller on a boat. While some bacteria have only a single flagellum, others, such as *E. coli*, have multiple flagella distributed over the cell surface. Rotating all their flagella in a counterclockwise direction allows the bacterium to swim—and it has been proposed that the clockwise movement of at least one flagellum will cause the bacterium cell to stop swimming and start tumbling.

*E. coli* is able to control the time it spends swimming or tumbling to move towards a nutrient, such as glucose, or away from certain harmful chemicals. However, the details of how the number of flagella and the direction of rotation (clockwise or counterclockwise) influence the motion of the bacterium are not fully understood.

Now, Mears et al. have used 'optical tweezers' to immobilize individual *E. coli* cells under a microscope, and then track both their swimming behavior and the movements of their flagella. This revealed that the individual flagella on the same cell tend to move in a coordinated way. Therefore, whilst tumbling could be caused by a single flagellum stopping swimming behavior, it often involved a concerted effort by many of the cell's flagella.

After observing that *E. coli* cells with more flagella spent less time tumbling than would be predicted if a single flagella always 'vetoed' swimming, Mears et al. propose a new mathematical relationship between the number of flagella on the cell, the direction of rotation, and the resulting probability that the cell will tumble. This work shows that swimming behavior in bacteria is less affected by variations in the number of flagella than previously thought—and this phenomenon may provide evolutionary advantages to *E. coli*. The next step is to explore the mechanism by which bacteria coordinate their flagella.

run/tumble swimming behavior of the whole, multi-flagellated cell remains poorly understood. The number of flagella on an individual swimming cell can vary greatly, from one to more than ten (*Cohen-Ben-Lulu et al., 2008*) (*Figure 1—figure supplement 1*), yet very few studies are available to indicate how flagellar number affects swimming behavior. The only direct measurements of flagellar dynamics in swimming cells have been limited to short durations (~1 s) (*Turner et al., 2000*; *Darnton et al., 2007*). The absence of long-term observations has precluded the development of a detailed mapping between flagellar state and cell swimming behavior. As a result, most theoretical models of bacterial chemotaxis are limited to treating an individual motor, or simply assume that all cells have a single flagellum (*Bray et al., 2007*; *Kalinin et al., 2009*; *Matthaus et al., 2009*; *Jiang et al., 2010*; *Flores et al., 2012*). Quantifying the mapping from single-flagellum state to whole-cell swimming behavior thus remains a missing link to developing an end-to-end picture of bacterial chemotaxis.

A number of theoretical models have been put forward in an attempt to describe this mapping. One such model invokes a 'voting' mechanism, in which cells tumble only if a majority of flagella rotate CW (*Ishihara et al., 1983*; *Spiro et al., 1997*; *Andrews et al., 2006*; *Vladimirov et al., 2008*; *Jiang et al., 2010*). However, by observing fluorescently labeled flagella during individual tumbles, Turner et al. established that CW rotation of a single flagellum is sufficient to 'veto' a run (*Turner et al., 2000*) (*Figure 1A*). Refined versions of this 'veto model' were recently developed (*Vladimirov et al., 2010*; *Sneddon et al., 2012*), based on careful, slow-motion observations of tumbles (*Darnton et al., 2007*). However, the extent to which these details are relevant for modeling swimming behavior is unknown, because no measurements have directly correlated long-term swimming behavior with flagellar activity in the same cell.

In this study, we present simultaneous, prolonged observations of individual flagella and whole-cell swimming. Using an optical trap to hold a swimming cell (*Min et al., 2009*) while simultaneously imaging its fluorescently labeled flagella, we relate directly the number and state of each flagellum on a cell to its swimming behavior. Our measurements establish a simple mapping between CCW/CW flagellar

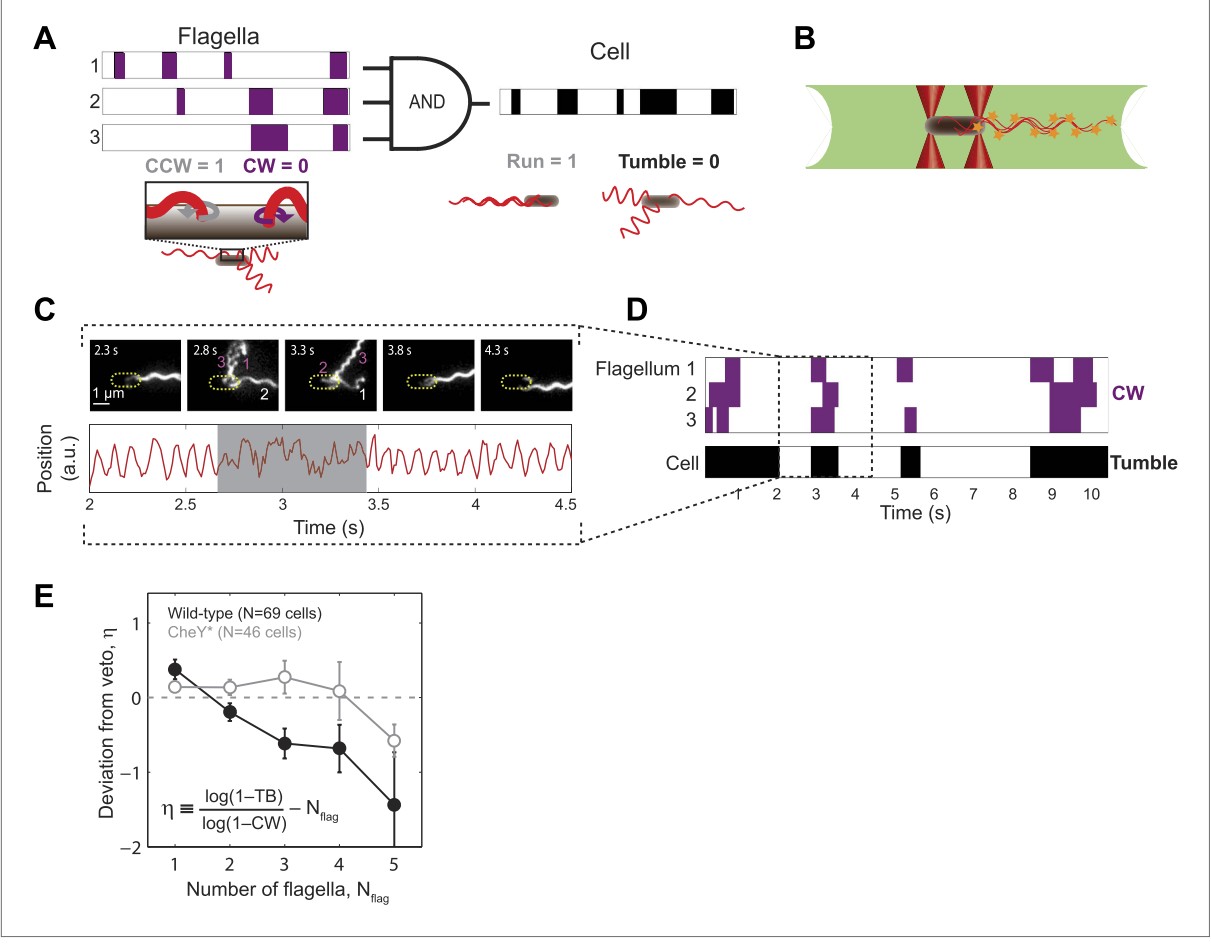

**Figure 1**. Wild-type *E. coli* cells deviate from the 'veto' model. (**A**) The mapping relating the run/tumble state of the cell to the CCW/CW state of its flagella according to the veto model. Schematic time trace from a cell with 3 flagella, showing CW (purple) and CCW (white) intervals for each flagellar motor and the resulting tumbles (black) and runs (white) of the cell. The veto model corresponds to an AND gate, by which cell runs only occur when all flagella rotate CCW (where CCW = 1, CW = 0, run = 1, and tumble = 0). (**B**) Schematic of a cell held by two optical traps (red cones) in the fluorescence excitation volume (green) within the sample chamber. (**C**) Representative data trace from a trapped cell with three flagella. Still images of fluorescently labeled flagella at 0.5-s intervals (top). The approximate location of the unlabeled cell body is indicated by a dashed yellow line. Flagella rotating CW (purple) and CCW (white) are numbered in frames in which they appear distinct. Corresponding cell-body rotation signal for the same cell (red line, bottom) as detected from deflections of the trapping laser. Tumbles (shaded area) were determined from the erratic cell-body rotation signal. (**D**) Long-term time trace of CCW/CW flagellar rotation state and run/tumble cell swimming state. CW intervals (purple, top) for each flagellum were determined from the fluorescence images. Tumbles (black, bottom) were determined from the cell-body rotation signal. (**E**) Mean deviation $\eta$ from the veto model vs number of flagella per cell. Wild-type cells (solid black circles) with multiple flagella deviate significantly from the model (p=0.0003, N = 69 cells). CheY* cells (open gray circles; N = 46 cells) match the model (p=0.77). Error bars denote SEM. See 'Materials and methods' for more details.

The following figure supplements are available for figure 1:

**Figure supplement 1**. Distribution of flagellar number.

**Figure supplement 2**. Instrument layout.

**Figure supplement 3**. Laser temporal interlacing scheme.

**Figure supplement 4**. Sample data from representative cells.

**Figure supplement 5**. Cells with flagella in the *curly-1* state rarely run.

rotation and run/tumble bacterial swimming state for cells with arbitrary numbers of flagella. Surprisingly, we find that *E. coli* cells wild-type for chemotaxis do not strictly comply with the veto model, because the states of individual flagella in the same cell are strongly coupled. Flagella do not switch independently, and as a result, tumbles typically involve many of the flagella on the cell. The behavior of multi-flagellated cells can still be mapped to the simple veto model by renormalizing the flagellar number to a lower effective number of independent flagella. As for the cause of inter-flagella coupling, our data strongly favor a mechanism involving fluctuations of the signaling network rather than direct, physical interactions between flagella. This model is supported by the observation that mutant cells in which flagellar switching is decoupled from the chemotaxis network do obey the simple veto model, as well as by stochastic simulations of the swimming behavior.

## Results

### The tumble bias of wild-type cells deviates from the predictions of the veto model

According to the veto model, a cell tumbles whenever any one flagellum rotates CW (*Figure 1A*). Thus, the probability that a cell runs equals the probability that all of its flagella remain CCW. As a consequence, cells with more flagella are expected to tumble more, since there is a higher chance that at least one flagellum will deviate from the consensus and 'veto' the run. These predictions can be stated mathematically, under the assumption that the rotational direction of each flagellum is independent of the other flagella. In a cell with $N_{flag}$ flagella, the average tumble bias *TB*—the fraction of time a cell spends tumbling—will be given by

$$TB = 1 - (1 - CB)^{N_{flag}} \qquad (1)$$

where *CB* is the average clockwise bias—the fraction of time the cell's flagella rotate CW ('Materials and methods').

To test this prediction, we quantified the swimming behavior of individual *E. coli* cells wild-type for chemotaxis (strain HCB1660 [*Turner et al., 2010*]; see 'Materials and methods', *Table 1*) using an instrument combining optical tweezers and epi-fluorescence imaging (*Figure 1B* and *Figure 1—figure supplement 2*). The instrument allowed us to measure simultaneously run/tumble behavior and flagellar dynamics in the same cell. The optical trap was used to hold each end of a single cell in place and the light scattered by the cell was utilized to monitor its swimming behavior, as described previously (*Min et al., 2009*, *2012*). As shown in *Figure 1C* (*Video 1*), cell runs were identified from oscillatory time signals due to cell body rotation at a frequency of ~10 Hz. Cell tumbles were identified as periods of erratic motion during which the flagellar bundle was disrupted (*Min et al., 2009*). Flagella were fluorescently labeled using the method of *Turner et al. (2010)*. High speed, epi-fluorescent, stroboscopic imaging (*Turner et al., 2000*; *Figure 1—figure supplement 3*) was used to resolve individual flagella (*Figure 1C*). Since the trapped cell remained in the field of view for a prolonged period, flagella were observed through multiple runs and tumbles (typically ~5 events), limited by the time until flagella became too dim to discern due to photobleaching (~8 to 40 s). The rotational direction of each flagellum was determined by observing its shape during 100-ms time windows. As shown by *Darnton et al. (2007)*, flagella may take on different helical waveforms depending on their rotational state. These waveforms, termed '*normal*', '*semi-coiled*', '*curly-1*', and '*curly-2*', can be visually identified based on their pitch and wavelength (*Figure 1C*, *Figure 1—figure supplement 4*). CCW rotating flagella were identified based on the *normal* conformation, which they have been shown to adopt exclusively (*Darnton et al., 2007*), while CW rotating flagella were identified by their *curly-1* or *semi-coiled* shape. From the identification of CCW and CW intervals, the cell's mean CW bias was determined by averaging the fraction of time that all the flagella on the cell spent CW ('Materials and methods').

Our assay allowed us to determine all the parameters in *Equation (1)* directly. For each cell, we measured the tumble bias (using the optical trap), flagellar number $N_{flag}$, and CW bias (using fluorescence imaging). We used these values to compare our experimental data to the prediction of the veto model. Reorganizing *Equation (1)*, we define the parameter $\eta$:

$$\eta \equiv \frac{\log(1 - TB)}{\log(1 - CB)} - N_{flag} \qquad (2)$$

**Table 1.** Strains and plasmids used in this work

| Strain | Genotype | Comments | Source |
|---|---|---|---|
| HCB1660 | *fliC*::Tn5 (Kan$^R$) | 'wild type' Contains plasmid pBAD33-fliC$^{S219C}$ | (*Turner et al., 2010*) Gift of H Berg |
| PM87 | *cheBYZ*::FRT, *fliC*::Tn5 (Kan$^R$) | 'CheY*' Contains plasmids pMS164 and pPM5 | This study |
| RP437 | | Wild-type for chemotaxis | (*Parkinson and Houts, 1982*) |
| SK109 | *cheBYZ*::Cm | | This study |
| SK110 | *cheBYZ*::FRT | | This study |
| SK112 | *cheBYZ*::FRT, *fliC*::Tn5 (Kan$^R$) | | This study |
| Plasmids | | | |
| pBAD33 fliC$^{S219C}$ | *fliC$^{S219C}$* under P$_{araBAD}$ promoter, Cm$^R$, p15a origin | Expresses mutant version of FliC for fluorescent labeling | (*Turner et al., 2010*) Gift of H Berg |
| pPM5 | *fliC$^{S219C}$* under P$_{araBAD}$ promoter, Amp$^R$, colE1 origin | Expresses mutant version of FliC for fluorescent labeling | This study |
| pMS164 | *cheY$^{D13K}$* under P$_{lacOP}$ promoter, Cm$^R$, pSC101 origin | Expresses constitutively active version of CheY | (*Alon et al., 1998*) Gift of P Cluzel |
| pDK46 | | Helper plasmid | (*Datsenko and Wanner, 2000*) |
| pKD3 | | Template for Cm$^R$ cassette | (*Datsenko and Wanner, 2000*) |
| pCP20 | | Helper plasmid | (*Cherepanov and Wackernagel, 1995*) |

which quantifies the deviation of the data from the veto model. Comparing *Equation (2)* to *Equation (1)*, $\eta$ may also be interpreted as the difference between two terms: the number of flagella estimated from the veto model based on the cell's swimming behavior, and the number of flagella on the cell as determined by counting directly. *Figure 1E* (black circles) shows $\eta$ against the flagellar number $N_{flag}$. $\eta$ was calculated for each individual cell and then averaged over all cells with a given number of flagella. Unexpectedly, we found that wild-type cells with multiple flagella systematically deviated from the predicted behavior. Specifically, $\eta$ was consistently negative for cells with $N_{flag} > 1$ (35/48 cells), indicating that cells with multiple flagella tumbled less than expected from the model. In the context of the veto model, the cells behaved as if they had a smaller number of flagella than what they actually had.

We first considered the possibility that a more detailed version of the veto model might explain the observed behavior and reconcile this discrepancy. A recent study by *Sneddon et al. (2012)* used the observations of *Darnton et al. (2007)* to refine the veto model. Specifically, the Sneddon model states that a cell with a minimum of $X$ CCW flagella will run rather than tumble, provided the remaining ($N_{flag}–X$) CW flagella are in the *curly-1* conformation only ($X$ is a parameter in the model, with possible values in the range [1, $N_{flag} − 1$]). Thus, the simple veto model considered above corresponds to $X = N_{flag}$, and the least perturbative refinement to the model corresponds to $X = N_{flag} − 1$, in which a cell with a single *curly-1* flagellum still runs. However, in our measurements we observed that cells with a single CW flagellum in the *curly-1* state still tumbled 82% of the time (44 s of cumulative time in which one flagellum was in the *curly-1* state; *Figure 1—figure supplement 5*). Modifying the Sneddon model to allow runs 18% of the time was not sufficient to reproduce the trend observed in *Figure 1D*, *Figure 1— figure supplement 5*.

## Wild-type tumbles typically involve multiple flagella

To investigate the discrepancy between our data and the veto model, we next examined individual tumble events in greater detail. In agreement with the original observation of *Turner et al. (2000)*, we found that CW rotation of a single flagellum was indeed sufficient to cause a tumble in multi-flagellated cells (*Figure 1—figure supplement 4*, samples A, D and E). However, we also observed that more

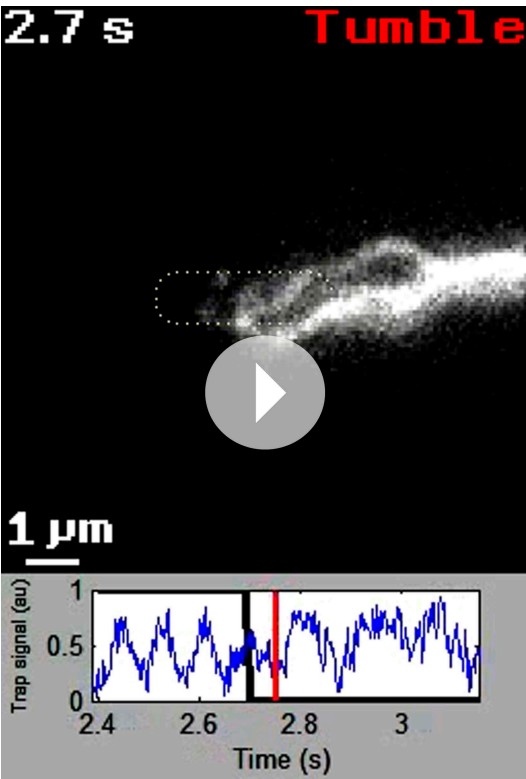

**Video 1**. **Video of trapped wild-type cell with three labeled flagella as it runs and tumbles.** Slow motion video of the wild-type cell in *Figure 1C* with three long, fluorescently labeled flagella. The approximate location of the unlabeled cell body is indicated by the dotted line. The trap signal used to determine runs and tumbles (bottom, scrolling blue curve) measures the position of the cell body in the trap as it rotates. At the beginning of the video (time stamp = 2.4 s), all three flagella are in a bundle and the cell is running. One by one, the flagella switch to CW rotation (2.7–3.0 s), which disrupts the bundle and causes the cell to tumble. Flagella can be observed in all three waveforms, *normal, semi-coiled,* and *curly-1*. Near the end of the video, the flagella all return to CCW rotation and coalesce into a bundle, causing the cell to resume running (3.4 s). In addition to the three long flagella, a short flagellar stub is visible. The stub does not affect the swimming behavior and is not analyzed. Scale bar in bottom left corner is 1 µm. Frames were recorded at 400 frames per second, video shows every other frame at 20 frames per second.

than half of tumbles in multi-flagellated cells (56%, 117/210 events) actually involved multiple CW flagella. *Figure 2A* shows a representative trace from a wild-type cell with three flagella. There are times during each tumble in the trace where all three flagella are in a CW state. As shown in *Figure 2C*, the number of CW flagella 'participating' in a tumble (black circles) was significantly larger than would be expected if flagella were independently switching (gray dashed line, obtained from simulations of a cell with independent flagella; see 'Materials and methods'). Our results thus suggest that while a single CW flagellum is sufficient to induce a tumble (in agreement with a simple veto model), flagella are also coupled and may thus switch in groups, in a correlated fashion. Further evidence for inter-flagella coupling was obtained by calculating the cross-correlation between pairs of flagella on a given cell. We found a significant correlation between the rotational directions of pairs of flagella on the same cell (*Figure 2D*, black data points). This correlation persisted for ~1 s, the average duration of a tumble. Our findings are consistent with previous observations by Terasawa et al. on surface-immobilized cells (*Terasawa et al., 2011*). There, correlations between individual motors on the same cell were detected by monitoring beads attached to flagellar stubs, as opposed to complete flagella on swimming cells in our present work.

The source of inter-flagellar correlation remains under debate. Terasawa et al. observed that mutant cells, in which the concentration of CheY-P was decoupled from the chemotaxis network, displayed no correlations (*Terasawa et al., 2011*). This led us to likewise investigate the behavior of a strain, PM87, expressing a constitutively-active CheY (CheY$^{D13K}$ [*Alon et al., 1998*], denoted CheY*, see *Table 1* and *2*). The protein was exogenously expressed, with the expression level chosen such that the population-averaged tumble bias matched that of wild-type cells ('Materials and methods'). A representative trace from a CheY* cell with three flagella is shown in *Figure 2B* (see also *Figure 1—figure supplement 4*, and *Video 2*). Upon inspection, flagellar switching appears far less correlated than in wild-type cells (*Figure 2D*, compare black and gray data points). Comparing

54 wild-type and 24 CheY* cells with the same mean CW biases (0.11 ± 0.07 vs 0.11 ± 0.07, mean ± SD) we found that, on average, fewer CW rotating flagella participated in tumbles in the CheY* strain (*Figure 2C*, open circles). Moreover, the number of participating flagella in the CheY* strain closely matched the expectation for cells with independently switching flagella (*Figure 2C*, dashed line). (This number deviates from unity and trends upwards with number of flagella simply because of the finite probability that two tumbles overlap by chance.) Our results indicate that when the signal for flagellar motors to switch their rotational state is decoupled from the chemotaxis network, the motors switch independently. Based on our interpretation of the wild-type data, we thus expect CheY* cells to

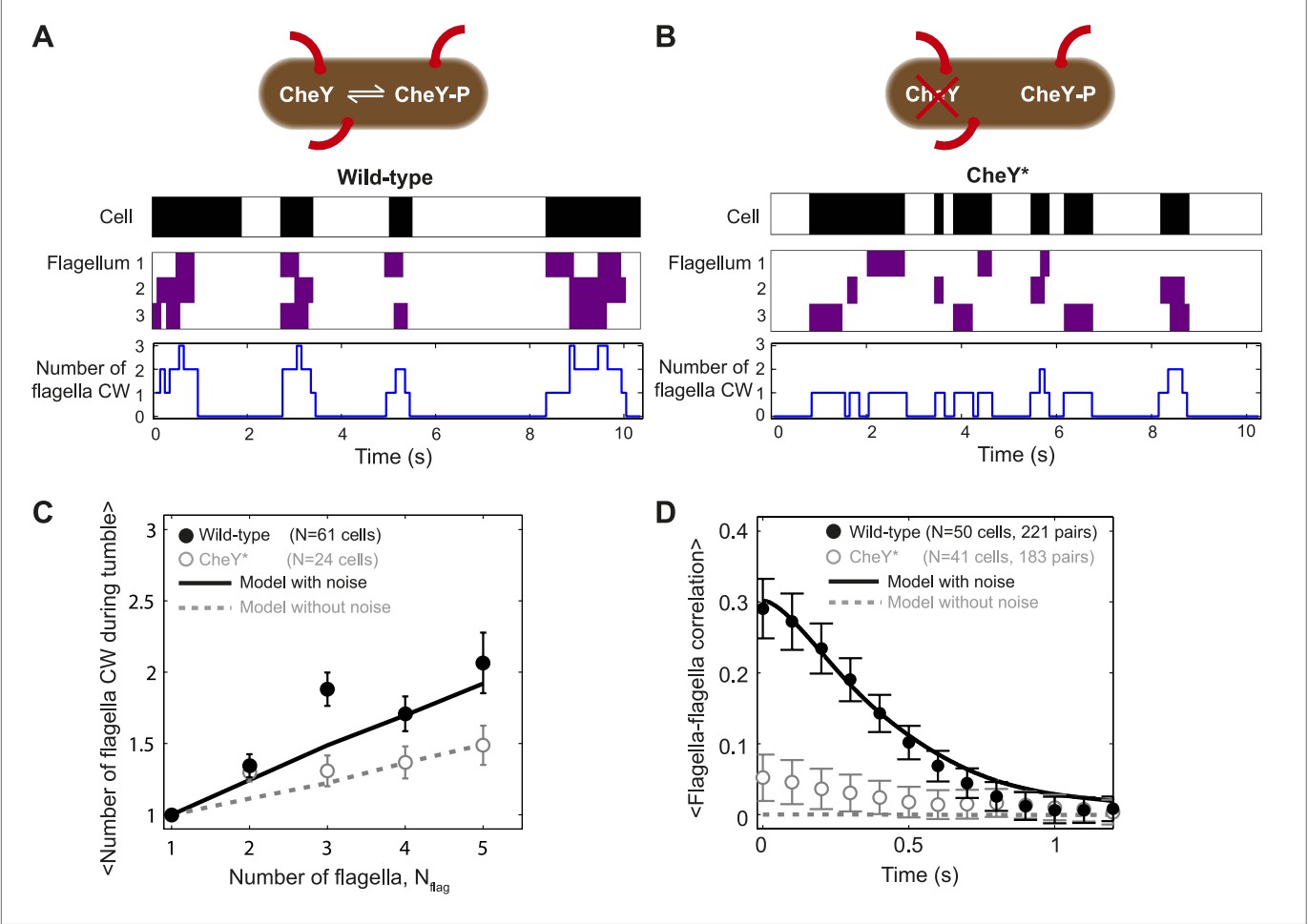

**Figure 2**. Tumbles in wild-type cells involve multiple CW flagella. (**A**) A typical time trace from a wild-type cell with 3 flagella. Colors indicate runs/tumbles (white/black, top) and CCW/CW (white/purple, middle). The blue line (bottom) shows the corresponding number of CW flagella at each time point. (**B**) Same as (**A**) for a typical CheY* cell with 3 flagella. (**C**) Mean of the maximum number of CW flagella during a tumble vs number of flagella per cell. Consistently more flagella are CW during tumbles in the wild-type (black circles; $N$ = 61 cells) compared to the CheY* strain (open gray circles; $N$ = 24 cells). Simulations incorporating fluctuations in CheY-P (black line) and without fluctuations (gray dashed lines) reproduce the observed trends (simulations detailed in the text and **Figure 4**). (**D**) Cross-correlation between flagella pairs, averaged over all pairs and all cells. Wild-type (black circles) match simulations with fluctuations in CheY-P (black line). CheY* strain (open gray circles) matches simulations without fluctuations in CheY-P (gray dashed line), which exhibit almost no correlation. Error bars denote SEM. See 'Materials and methods' for more details.

The following figure supplements are available for figure 2:

**Figure supplement 1**. Flagellar transition rates vs number of flagella per cell.

**Figure supplement 2**. Flagellar transition rates vs number of flagella per cell.

adhere to the simple veto model. As shown in the plot of $\eta$ in **Figure 1D** (open circles), CheY* cells indeed match the veto model closely ($\eta$ = −0.08 ± 0.15, mean ± SEM). The existence of correlations between flagella states in wild-type cells may thus explain why cells with multiple flagella deviate from the veto model.

## Wild-type behavior can be described by a veto model with a lower effective number of flagella

Our results so far suggest that, while wild-type cells obey the fundamental premise of the veto model—that is, a single CW flagellum is sufficient to induce a tumble—the presence of inter-flagella correlations leads to the failure of **Equation (1)** in relating the observed CW bias and tumble bias. To describe the

**Table 2.** Primers used in this work

| Primer | Sequence | Comments |
|---|---|---|
| SK140F | TGCGTGGTCAGACGGTGTATGCGCTAAGTAAGGATTAACG GTGTAGGCTGGAGCTGCTTC | *cheBYZ* deletion forward |
| SK140R | GCCTGATATGACGTGGTCACGCCACATCAGGCAATACAAA CATATGAATATCCTCCTTAG | *cheBYZ* deletion reverse |
| SK141F | CCTTAAACCCGACGGATTGC | *cheBYZ* deletion check forward |
| SK141R | TTGCTGCCACACATCAAGC | *cheBYZ* deletion check reverse |
| SK163F | AGGGTTATTGTCTCATGAGC | pZE11 sequencing forward |
| SK163R | GTTTTATTTGATGCCTCTAG | pZE11 sequencing reverse |
| PM7F | GGG GACGTC ATCGATGCATAATGTGCCTG | amplify $P_{araBAD}$ $fliC^{S219C}$ forward |
| PM7R | GGG GTCGAC TTAACCCTGCAGC | amplify $P_{araBAD}$ $fliC^{S219C}$ reverse |

relation between single flagellar state and whole-cell behavior successfully, this expression must then be modified to account for flagellar correlations. To this end, we examined the relation between CW bias and tumble bias in all individual cells having a given flagellar number (***Figure 3A,B***). ***Equation (1)*** defines a single curve, along which the CW bias and tumble bias of all cells with $N_{flag}$ flagella should lie (dashed line in ***Figure 3A,B***). As expected, the CheY* cells follow these predicted curves closely for all $N_{flag}$ values (***Figure 3B***) ($R^2$ = 0.89). In contrast, wild-type cells with multiple flagella consistently fall below the predicted curves (***Figure 3A***) (35/48 cells).

Based on our observation that wild-type cells exhibited $\eta$ values consistent with cells with a lower number of flagella than the actual value, we hypothesized that wild-type behavior may be described within the framework of the veto model by allowing the parameter $N_{flag}$ in ***Equation (1)*** to deviate from the actual flagellar number. As shown in ***Figure 3A*** (solid lines), using the flagellar number as a fitting parameter (now denoted $N_{eff}$) indeed allows for a good match for the wild-type data ($R^2$ = 0.85). In this revised equation, $N_{eff}$ can be thought of as the 'effective' number of independent flagella on a cell, which captures the fact that flagella in wild-type cells switch in a correlated manner. Consistent with this picture, the effective number of flagella $N_{eff}$ is consistently smaller than the actual flagellar number $N_{flag}$ (***Figure 3C***, black circles; $N_{eff}$ can be approximated by $N_{eff} = 1.27 \times N_{flag}^{0.5}$, see ***Figure 3—figure supplement 1***). As a control, estimating $N_{eff}$ for CheY* cells produces values very close to the original flagellar number $N_{flag}$ (***Figure 3B***, solid line and ***Figure 3C***, open circles). The introduction of the parameter $N_{eff}$ allows us to formulate a generalized veto model, which describes the mapping between the CW bias and tumble bias for both wild-type and CheY* cells. The generalized model now defines a universal curve,

$$1 - TB = (1 - CB)^{N_{eff}} \qquad (3)$$

along which all individual cells of both genotypes should lie (using $N_{eff} = N_{flag}$ for CheY* strain and the best fit value of $N_{eff}$ for wild-type). As seen in ***Figure 3D***, this expression successfully collapses all single-cell data from both strains and all flagellar numbers.

## A theoretical model incorporating CheY-P fluctuations reproduces wild-type swimming behavior

Our results show that *E. coli* cells adhere to the veto model, but that inter-flagellar correlations lead to a renormalization of the effect of flagellar number. What is the source of these flagellar correlations? The absence of correlations in the CheY* strain and its adherence to a simple veto model provide an important clue to understanding the mechanism of inter-flagellar coupling. In wild-type cells, CheY-P levels are subject to phosphorylation and de-phosphorylation reactions by chemotaxis network components and are believed to fluctuate in time (***Korobkova et al., 2006***; ***Sneddon et al., 2012***). In contrast, in the mutant strain, CheY* levels are decoupled from the network and are thus expected to be constant over the timescales of interest (***Korobkova et al., 2006***; ***Min et al., 2009***). Terasawa et al. proposed that fluctuations in CheY-P levels may thus provide a mechanism by which the CW biases of multiple flagella on a cell can be coupled (***Terasawa et al., 2011***). To test whether such a mechanism could account for the different features of bacterial swimming observed in our study, we performed simulations of whole-cell swimming driven by the chemotaxis network. In particular, we investigated

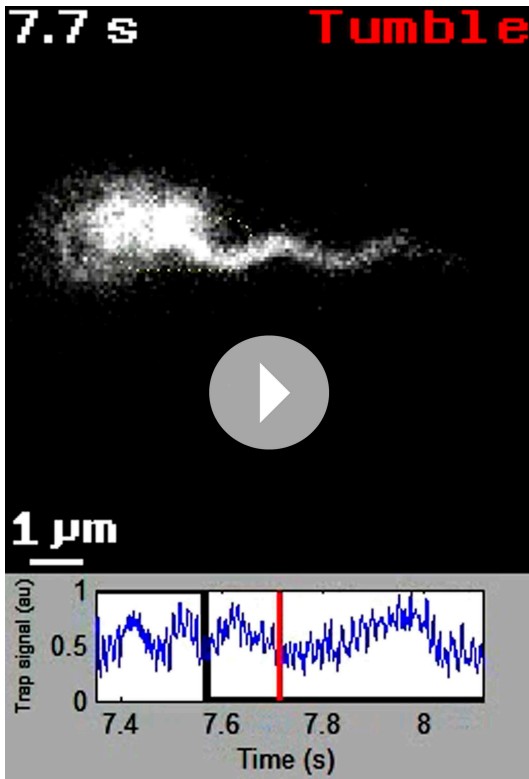

**Video 2**. **Video of trapped CheY\* cell with two labeled flagella as it runs and tumbles.** Slow motion video, similar to *Video 1*, of a trapped CheY\* cell with two fluorescently labeled flagella. Still images from this cell are shown in *Figure 1—figure supplement 4*, sample D. The approximate location of the unlabeled cell body is indicated by the dotted line. At the beginning of the video (time stamp = 7.4 s), both flagella are in a bundle, rotating CCW in the *normal* waveform, and the cell is running. At 7.6 s, one flagellum switches to CW rotation and transitions to the *semi-coiled* waveform, which disrupts the bundle and causes the cell to tumble. At the 8.5 s, the *semi-coiled* flagellum returns to CCW rotation and both flagella coalesce into a bundle, causing the cell to resume running. Scale bar in bottom left corner is 1 µm. Frames were recorded at 100 frames per second, video plays at 20 frames per second.

how fluctuations in CheY-P concentration could produce differences in inter-flagellar correlations between wild-type and CheY\* cells and consequent differences in their respective mappings of CW bias to tumble bias.

We performed stochastic simulations of the chemotaxis network and resulting flagellar motor activity and then applied the veto rule that CW rotation of a single flagellum leads to cell tumbling (*Figure 4A,B*; 'Materials and methods'). For wild-type cells, we incorporated fluctuations in CheY-P concentration using the approach of *Sneddon et al. (2012)*. The model was constrained using transition rates between flagella waveforms that were directly extracted from our experiments (*Figure 4—figure supplement 1* and *Tables 3 and 4*). We found that, consistently, simulations with CheY-P fluctuations were able to duplicate the observed wild-type behavior, while simulations without fluctuations matched CheY\* data and the predictions of the 'naïve' veto model. Specifically, using only two parameters—the amplitude and the characteristic timescale for CheY-P fluctuations (*Table 3*)—as fitting parameters, our simulations simultaneously reproduced (i) the relation between flagellar number and flagellar participation in tumbles (*Figure 2C*); (ii) temporal correlations between flagella (*Figure 2D*); (iii) the effective number of flagella in multi-flagellated wild-type cells (*Figure 3C*); and (iv) the degree of deviation from the veto model (*Figure 4C*).

## Discussion

The experimental approach described here allows the simultaneous, long-term observation of flagellar activity and swimming behavior in a single cell. By imaging many individual tumbles (*N* = 203 in wild-type cells), we are able to describe in great detail the underlying structure of a tumbling event. For instance, we can follow the sequence of flagellar waveforms that occurs when motors switch from CCW to CW rotation and back to CCW. *Figure 4—figure supplement 2* shows the distribution of possible sequences of flagellar waveforms during tumbles. In particular, the sequence of states from *normal* to *semi-coiled* to *curly-1* that we observed was described by *Darnton et al. (2007)* as a 'canonical tumble'. Although we cannot rule out that runs and tumbles in the optical trap are different in some ways than those in free swimming cells (*Min et al., 2009*), our results are in qualitative agreement with these previous observations.

Our measurements also reveal the relationship between the cell's run/tumble state and the CCW/CW rotational state of its flagella. In a multi-flagellated wild-type cell, a single CW flagellum (either *semi-coiled* or *curly-1*) is sufficient to induce a tumble, in agreement with the simple veto model (*Turner et al., 2000*; *Darnton et al., 2007*). However, the number of CW flagella during a tumble typically exceeds that expected from a cell with independently switching flagella (*Figure 2C*). The high fraction of CW flagella during tumbles in wild-type cells is in qualitative agreement with previous measurements by *Turner et al. (2000)*, who observed that a majority of tumbles involved multiple flagella leaving the bundle. Our measurements using the CheY\* strain provide an important piece of evidence

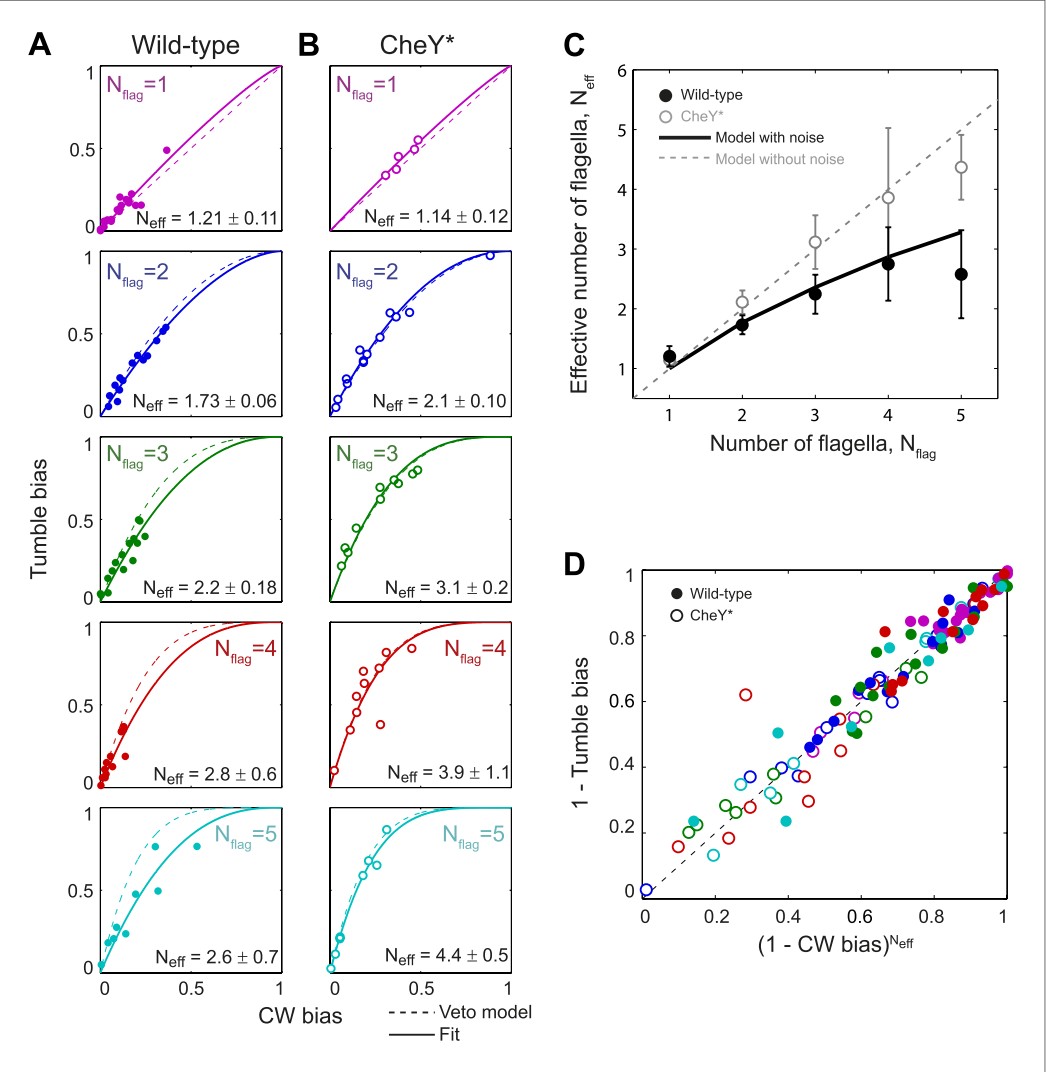

**Figure 3**. Wild-type behavior matches the veto model for cells with a lower effective number of flagella. (**A**) Tumble bias vs CW bias for individual wild-type cells (*N* = 69), plotted separately for different numbers of flagella per cell ($N_{flag}$ = 1, purple; 2, blue; 3, green; 4, red; 5, cyan). The prediction from the veto model in ***Equation (1)*** (dashed lines) does not match the data for cells with multiple flagella ($R^2$ = 0.88, 0.60, 0.41, 0.39 for $N_{flag}$ = 2, 3, 4, 5). The data were fit (solid lines) to ***Equation (1)***, while allowing the number of flagella to be used as a fitting parameter, $N_{eff}$. Error bars denote SD. (**B**) Same as (**A**) for CheY* (open circles, same color code as [**A**] *N* = 46 cells). The veto model prediction (dashed lines) matches the data well ($R^2$ = 0.91, 0.97, 0.93, 0.67, 0.98 for $N_{flag}$ = 1, 2, 3, 4, 5). Fits (solid lines) yield $N_{eff}$ values almost identical to $N_{flag}$. (**C**) Fitted $N_{eff}$ values vs number of flagella per cell for wild-type (black circles) and CheY* (open gray circles) cells. Simulations (described in the text) reproduce the observed trends. (**D**) Data points from individual wild-type (solid circles) and CheY* (open circles) cells all collapse onto a single line when using $N_{eff}$ from fits to wild-type data in (**A**) and the actual flagellar number $N_{flag}$ for CheY* cells in (**B**). Error bars denote SEM. See 'Materials and methods' for more details.

The following figure supplements are available for figure 3:

**Figure supplement 1**. Fit to $N_{eff}$ vs $Nf_{lag}$.

linking inter-flagellar coupling to the chemotaxis network. We propose that fluctuations in the concentration of CheY-P are at the heart of wild-type *E. coli* behavior. In our theoretical analysis, the existence of temporal fluctuations was sufficient to explain all of our data. Stochastic simulations with and without CheY-P fluctuations (representing wild-type and CheY* cells, respectively) reproduced all of the observed differences between our two strains. ***Figure 4A*** summarizes how CheY-P fluctuations could lead to

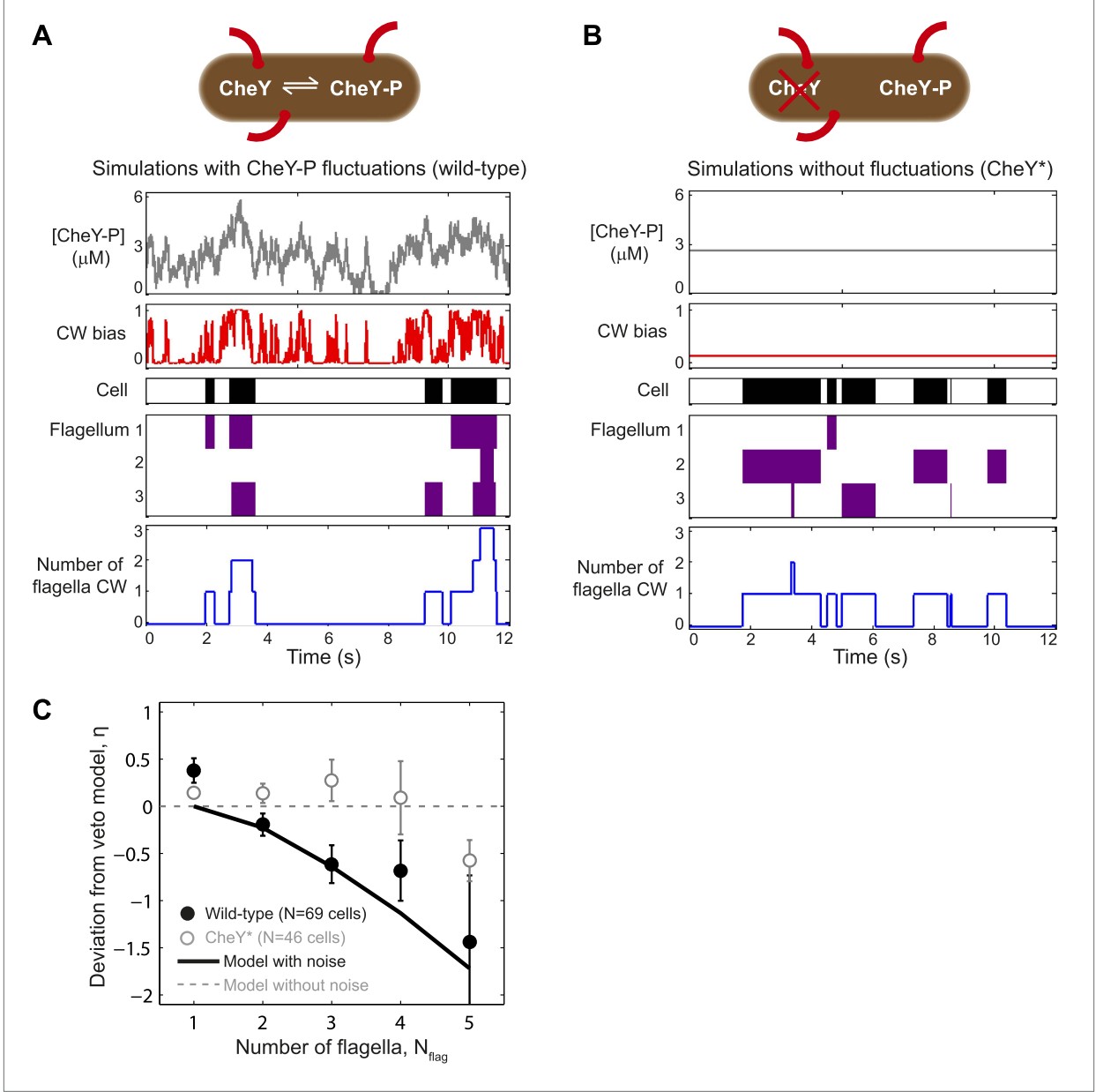

**Figure 4**. A theoretical model incorporating CheY-P fluctuations reproduces wild-type data. (**A**) Simulated time trace for a wild-type cell. Representative simulated time trace of CheY-P concentration (gray line, top), CW bias (red), run/tumble state (white/black), CCW/CW flagellar rotational direction (white/purple) and number of CW flagella (blue line, bottom) for a cell with 3 flagella. CheY-P simulation parameters are described in the text and in *Table 3*. (**B**) Same as (**A**) from a simulated CheY* cell, in which CheY-P concentration (gray line) does not fluctuate. (**C**) Deviation from veto model. The theoretical model that includes CheY-P concentration fluctuations (black line) reproduces the wild-type data (black circles). A simple veto model with constant CheY-P concentration (gray dashed line) reproduces the CheY* data (open gray circles). Error bars denote SEM. See 'Materials and methods' for more details.

The following figure supplements are available for figure 4:

**Figure supplement 1**. Flagellar waveform transition rates.

**Figure supplement 2**. Flagellar waveform transition sequences.

**Figure supplement 3**. Flagellar waveform transition sequences.

**Figure supplement 4**. Tumble bias vs number of flagella per cell.

**Table 3.** Model parameters

| Parameter | Description | Value | Source |
|---|---|---|---|
| $k_{ccw->cw}$ | Motor switching rate from CCW→CW | 0.26 s$^{-1}$ | Our data |
| $k_{cw->ccw}$ | Motor switching rate from CW→CCW | 1.7 s$^{-1}$ | Our data |
| CB | Average clock-wise bias of wild-type motors | 0.13 | Our data |
| <Y> | Mean concentration of CheY-P | 2.59 µM | Fit to our data |
| $\omega$ | Characteristic motor switching time | 0.5 s | Our data |
| $\lambda^{-1}$ | Transition rate from *semi-coiled* to *curly-1* state | 0.68 s$^{-1}$ | Our data |
| x | From the model in Sneddon et al.; number of flagella that must be *normal* for a run to occur (while other flagella are *curly-1*) (*Sneddon et al., 2012*) | $N_{flag}$ (variable in *Figure 1—figure supplement 5*) | Our data |
| $\sigma^2$ | Variance in [CheY-P] | 1.0 µM$^2$ | Fit to our data |
| $\tau$ | Characteristic time-scale of fluctuations in [CheY-P] | 0.2 s | Fit to our data |
| $K_d$ | Midpoint of CW bias vs CheY-P response curve | 3.1 µM | (*Cluzel et al., 2000*) |
| H | Hill coefficient for CW bias vs CheY-P response curve | 10.3 | (*Cluzel et al., 2000*) |
| dt | Simulation time steps | 0.001 s | – |

Note: $\omega \equiv \dfrac{CB}{k_{ccw \rightarrow cw}}$

correlated flagellar switching. A well-known feature of the chemotaxis network is the sigmoidal relation between CW bias and CheY-P concentration (*Cluzel et al., 2000*; *Yuan and Berg, 2013*). A consequence of this non-linearity is that the probability of CW rotation is highly sensitive, and can respond dramatically to fluctuations in CheY-P levels, provided their amplitude is sufficiently large. As shown in *Figure 4A*, when CheY-P concentration is high, the cell experiences a near-100% probability of CW rotation and multiple motors switch CW at approximately the same time. In contrast, when CheY-P concentration is low, the probability of any motor rotating CW is essentially zero. This mechanism can explain the elevated number of CW flagella involved in tumbles (*Figure 2C*) and correlation between flagella states (*Figure 2D*). By contrast, in simulations where CheY-P level was held constant, flagellar switching was not as correlated and the majority of tumbles involved only a single CW flagellum (*Figure 4B*).

Despite the success of this model in reproducing our data, we must acknowledge that there is no direct experimental evidence to-date for the CheY-P fluctuations depicted in *Figure 4A*. Fluctuations in CheY-P have been inferred from experimental observations of CW bias in tethered-bead assays (*Korobkova et al., 2004*). However, the fluctuations described in that study are different in their time scale and amplitude from what we found required to produce the observed correlations in flagellar rotational direction ('Materials and methods'). Future investigation will be essential to resolving this issue and will likely have to involve direct measurements of CheY-P temporal dynamics in individual cells. Such measurements are challenging, but the development of intra-cellular fluorescence sensors for kinase activity in the network (*Sourjik and Berg, 2002*) provides a promising approach.

As an alternative mechanism for the inter-flagellar correlations observed by *Terasawa et al. (2011)* (and in the present work), *Hu and Tu (2013)* proposed that hydrodynamic interactions between nearby

**Table 4.** Flagella waveform transition rates

| Initial | Final | | |
| | **Normal** | **Semi-coiled** | **Curly-1** |
|---|---|---|---|
| *Normal* | | 0.28 ± 0.03 s$^{-1}$ | 0.08 ± 0.01 s$^{-1}$ |
| *Semi-coiled* | 1.6 ± 0.2 s$^{-1}$ | | 0.54 ± 0.08 s$^{-1}$ |
| *Curly-1* | 1.8 ± 0.2 s$^{-1}$ | 0.04 ± 0.02 s$^{-1}$ | |

Transition rates between different flagellar waveforms: *normal* (CCW), *semi-coiled* and *curly-1* (both CW). Data from wild-type cells (*N* = 52 cells, 203 tumbles). Values are mean ± SEM.

flagella could also generate correlations in their rotational direction. One consequence of their model is that the flagellar switching rates in a cell with a single flagellum will be different than those in multi-flagellated cells. However, this prediction is not borne out by our data. In our experiments, the number of flagella per cell did not have a significant effect on the switching rates between CCW and CW states, nor on the switching rates between different flagellar waveforms (*Figure 2—figure supplements 1 and 2*). While we cannot rule out the presence of hydrodynamic interactions between flagella, these must satisfy the strict requirement that switching rates remain independent of flagellar number. In light of these constraints, we believe a mechanism in which chemotaxis network fluctuations engender inter-flagella correlations to be more plausible. For cells with $N_{flag} > 4$, we note that both strains appear to deviate from the generalized veto model (*Figure 3C*). It is possible that hydrodynamic effects must be taken in consideration in cells with many flagella. Hydrodynamic coupling in the Hu model leads to a lower $N_{eff}$, in the direction of the deviation. Alternatively, a mechanism as described by *Sneddon et al. (2012)*, in which cells with many flagella can run while some of its flagella rotate CW in the *curly-1* state, could lead to a similar deviation (*Figure 1—figure supplement 5*). Finally, we must consider the possibility that the apparent deviation is due to systematic experimental error, since it is more difficult to determine visually the state of each flagellum on cells with many flagella.

While a large number of studies have elucidated mathematical relationships between many of the components of the chemotaxis network (*Block et al., 1982*; *Cluzel et al., 2000*; *Sourjik and Berg, 2002*; *Shimizu et al., 2010*; *Yuan et al., 2012*), there have been few experimental studies devoted to the relationship between individual flagella and whole-cell swimming (*Turner et al., 2000*; *Darnton et al., 2007*). As a result, existing models of bacterial chemotaxis have made drastically different assumptions in order to describe the swimming behavior of the whole cell (*Bray et al., 2007*; *Jiang et al., 2010*; *Vladimirov et al., 2010*; *Sneddon et al., 2012*). To the best of our knowledge, the current study provides the first experimentally-derived mathematical relation between flagellar and whole-cell-swimming states.

We propose that the details of this mapping are crucial for fully understanding bacterial chemotaxis. Results from recent theoretical models suggest that the details of flagellar mechanics can have significant effects on chemotactic drift. Turner et al. observed that, on average, the angular change in swimming direction upon tumbling increases as a function of the number of flagella that leave the bundle (*Turner et al., 2000*). Vladimirov et al. showed that when this effect is incorporated into a theoretical model of bacterial chemotaxis, the chemotactic drift is nearly doubled (*Vladimirov et al., 2010*).

Our observations that multi-flagellated wild-type cells tumble significantly less than expected also implies that the cell's swimming behavior (and presumably its chemotactic response) is robust against variations in the number of flagella (*Figure 4—figure supplement 4*). We hypothesize that this phenomenon may confer evolutionary advantages, in light of the large fluctuations in flagellar numbers within a cell population (*Figure 1—figure supplement 1*). If cells with many flagella did not behave like cells with fewer flagella, then they would spend the majority of their time tumbling, a behavior that would inhibit chemotaxis. *E. coli* thus appears to have developed a mechanism to achieve similar tumble biases with a wide range of flagellar number.

## Materials and methods

### Microbiology

#### Cell preparation

Experiments were performed using two *E. coli* strains (*Table 1*). The strain referred to as 'wild-type' is HCB1660 ([*Turner et al., 2010*], gift of Howard Berg), *ΔfliC* expressing FliC$^{S219C}$ from a plasmid under the control of arabinose. The mutant protein FliC$^{S219C}$ was constructed to be specifically labeled with a maleimide functionalized fluorescent dye (*Turner et al., 2010*). The strain referred to as 'CheY*' is PM87 (constructed for this study, see below), *ΔfliC ΔcheBYZ* expressing FliC$^{S219C}$ and CheY$^{D13K}$ from separate plasmids under the control of arabinose and IPTG, respectively. The mutant protein CheY$^{D13K}$ is constitutively active (*Alon et al., 1998*), such that the CW bias was determined by the concentration of CheY$^{D13K}$, decoupled from the chemotaxis network.

For each experiment, cells were picked from a single colony on an agar plate and grown overnight in 1 ml tryptone broth (1% [wt/vol] Bacto tryptone and 0.8% [wt/vol] NaCl) (*Saini et al., 2008*; *Min et al., 2009*) shaking at 265 RPM at 30°C with appropriate antibiotics. The overnight culture was diluted 100-fold into 12-ml tryptone broth and grown, shaking at 265 RPM at 30°C for 4.5 hr (to OD$_{600}$ ~0.5) with appropriate inducers (500 μM arabinose and 50 μM IPTG). To visualize flagella, we used a fluorescence labeling

protocol developed by *Turner et al. (2010)*. The over-day culture was harvested, washed twice by slow centrifugation (1300×*g*, 10 min) and gently resuspended in 1 ml motility buffer (MB) (*Darnton et al., 2007*) (10 mM KPO$_4$ (pH 7.0), 70 mM NaCl and 0.1 mM EDTA) and then in 0.5 ml MB. Flagella were specifically labeled using Alexa Fluor 532 C$_5$ Maleimide (A-10255; Life Technologies, Carlsbad, California). 1 mg of dry dye was dissolved in 300 µl H$_2$0 by vortexing for 1 min. Aliquots containing 50 µl of dissolved dye were stored at −20°C. Cells in 500 µl MB were gently mixed with 5 µl of the dissolved dye and then incubated with slow rotation (~10 RPM) at room temperature in the dark for 90 min. The labeled culture was washed and gently resuspended in 1 ml MB. Finally, cells were diluted 20-fold into 1 ml trap motility buffer (TMB) (70 mM NaCl, 100 mM Tris-Cl, 2% [wt/vol] glucose, and an oxygen-scavenging system [80 µg ml$^{-1}$ glucose oxidase and 13 µg ml$^{-1}$ catalase]) (*Min et al., 2009*) and injected into the flow cell for trapping. At all stages, resuspension by pipetting was avoided to prevent shearing of the flagella (*Turner et al., 2000*).

## Construction of strains

Bacterial strains and plasmids used in this study are listed in *Table 1*. Oligonucleotides used for generating mutations and creating plasmids are listed in *Table 2*. The generalized transducing phage P1*vir* was used in all transductional crosses (*Thomason et al., 2007*). Chromosomal mutations were introduced using a standard λ Red recombination method as described by *Datsenko and Wanner (2000)*. All primers were purchased from Integrated DNA Technologies (Coralville, Iowa), all sequencing was performed by ACGT (Wheeling, Illinois).

Strain PM87 (referred to as the CheY* strain in the text) was created from RP437 (wild-type for chemotaxis) in the following manner. First, *cheBYZ* was replaced by a chloramphenicol resistance cassette with flanking FRT sites using primers SK140F and SK140R along with pKD3 as a template (*Datsenko and Wanner, 2000*). This deletion was then moved into a clean RP437 using P1 transduction (*Thomason et al., 2007*), to create SK109 (*cheBYZ::*Cm). The chloramphenicol resistance cassette was removed from SK109 using Flp recombinase expressed from pCP20 (*Cherepanov and Wackernagel, 1995*) to obtain strain SK110 (*cheBYZ::*FRT). Next, *fliC::*Tn5 from strain HCB1660 was moved into strain SK110 using P1 transduction to obtain strain SK112. Finally, strain PM87 was created by transforming strain SK112 with plasmids pPM5 and pMS164 to express FliC$^{S219C}$ and CheY$^{D13K}$, respectively.

Standard molecular cloning techniques were used to construct plasmids (*Sambrook and Russell, 2001*). Primers PM7F and PM7R were used to PCR amplify *fliC$^{S219C}$* and the P$_{araBAD}$ promoter from plasmid pBAD33-fliC$^{S219C}$ (gift of H Berg, [*Turner et al., 2010*]) and to add restriction sites for AatII and SalI. The PCR product was then ligated into pZE11 (*Lutz and Bujard, 1997*) after digesting both with restriction enzymes AatII and SalI, to obtain plasmid pPM5.

## Instrument design

### Optical traps

Experiments were performed using a dual optical trap instrument incorporating a custom flow cell and stroboscopic, epi-fluorescent imaging. The instrument design is shown in *Figure 1—figure supplement 2*. The optical trap was constructed following a previously described design (*Comstock et al., 2011*). A 5-W, 1064-nm diode-pumped solid-state laser (BL-106C, Spectra-Physics, Santa Clara, California) was used to produce two optical traps via timesharing, by intermittently deflecting the laser with an acousto-optic deflector, AOD (DTD-274HD6, IntraAction, Bellwood, Illinois). The separation between the two traps was controlled by modulating the two deflection angles of the beams emanating from the acousto-optic deflector via the RF signal frequency driving the AOD. The IR beams were tightly focused to generate two optical traps by a 60×, water-immersion (1.2 NA) microscope objective (Nikon, Tokyo, Japan). An identical objective lens collected transmitted light for position detection and bright-field imaging, as described in *Min et al. (2009)*. The flow chamber was positioned between the two objective lenses and was moveable relative to the two traps in all directions by a motorized three-axis translational stage (ESP301; Newport, Irvine, California). Cell motion was detected directly by the optical traps themselves, using back-focal plane interferometry, in which trap light scattered by an object relays the object's position relative to the trap in all three directions (*Gittes and Schmidt, 1998*). All devices and timing were controlled using custom LabVIEW (National Instruments, Austin, Texas) code (*Comstock et al., 2011*).

### Fluorescence imaging

Epi-fluorescence imaging of trapped cells was achieved by excitation with a 532-nm laser (TECGL-30, World Star Tech, Toronto, Canada) and collecting backwards emitted photons with an EMCCD camera

(iXon3 860 EMCCD, Andor, Belfast, Ireland). The beam size at the sample plane was approximately 20 microns in diameter. To obtain clear images of the flagella as they rotated at angular frequencies ~100 Hz, it was necessary to take short exposures (*Figure 1—figure supplement 3*). To this end, we used stroboscopic illumination similarly to previous experiments (*Darnton et al., 2007*). This was achieved by intermittently deflecting the 532-nm laser beam with a second acousto-optic modulator (AOM-802AF1, IntraAction), so as to generate short-duration excitation pulses (20 μs). The EMCCD recorded 128 × 128-pixel images (~10 × 10 μm) synchronized with each excitation pulse (see timing in *Figure 1—figure supplement 3*, images *Figure 1—figure supplement 4*). Additionally, we pulsed the IR trapping laser out of phase with the fluorescence excitation at a rate of 16 kHz, a technique which has been shown to significantly reduce photo-bleaching, with minimal consequence to the trapping (*Brau et al., 2006*; *Comstock et al., 2011*) (*Figure 1—figure supplement 3*). The results were high speed movies showing sharp images of all the flagella on a cell as it runs and tumbles for many seconds (typically >10 s), along with synchronous signal traces from the optical traps. Fluorescence movies were saved using Solis software (Andor), and subsequently analyzed manually using Matlab (Mathworks, Natick, Massachusetts). When played back in slow-motion, flagella could be reliably counted by eye during tumbles.

## Fluidics
Experiments were performed in a custom-built microfluidic chamber. See *Min et al. (2012)* for a detailed description of the microfluidic chamber. Glass coverslips (12-545-M, 24 × 60−1, ThermoFisher, Waltham, Massachusetts) were sonicated in acetone for 5 min and rinsed with deionized water. The flow channel pattern was cut out from Nescofilm (Karlan, Phoenix, Arizona) using a laser engraver (Versa Laser, Scottsdale, Arizona) and placed between two coverslips, one of which had custom-drilled holes (0.05-inch diameter) for inlets and outlets. The Nescofilm flow channel pattern was bonded to coverslips by melting on a hot plate for 4 min. The completed flow chamber was inserted into a custom metal frame where inlet and outlet tubing (ABW00001; Tygon, Saint-Gobain, Paris, France) were screwed on for a tight seal. The two channels of the flow chamber were continuously injected with appropriate buffers using a syringe pump (PHD2000; Harvard Apparatus, Holliston, Massachusetts) at a linear speed of 30 μm/s, which is approximately equal to the swimming speed of a healthy cell. The upper channel was injected with TMB (see above), while the bottom channel contained cells in TMB. Cells were trapped in the bottom channel and then moved into the upper channel for observation by displacing the motorized flow chamber relative to the traps.

## Electron microscopy
TEM images were recorded using the JEOL 2100 cryo-Transmission Electron Microscope (TEM) at the Frederick Seitz Materials Research Laboratory Central Facilities at the University of Illinois at Urbana–Champaign, following the protocol of *Saini et al. (2010)*. Briefly, cells were grown as described above, and used without fluorescent labeling. Cells were fixed with glutaraldehyde and then placed on 200 Mesh Carbon Coated Copper grids (Cat. # 182; Canemco, Lakefield, Canada), which were used as supports for sample loading and imaging. Images were taken at 200 kV with a camera exposure lasting 1 s. Finally, images were contrast adjusted and an image dilation was performed in Matlab to make flagella more visible. The distribution of flagella per cell is shown in *Figure 1—figure supplement 1*.

## Data analysis
### Wavelet analysis for run-tumble detection
Determination of runs and tumbles from the optical trap signal was done as described previously (*Min et al., 2009*). See example trace in *Figure 1C*.

### Image analysis
Images were contrast adjusted to make flagella easier to see using Matlab. Movies were then manually analyzed by eye to count flagella and to identify flagella waveforms. The state of each flagellum was identified during each 100-ms time window (10 movie frames at 100 f.p.s.). See example cells in *Figure 1C* and *Figure 1—figure supplement 4*.

## Theoretical modeling
### The veto model
The veto model assumes that all flagella must be rotating counter-clockwise (CCW) for the cell to run. Any flagellum can 'veto' the run by rotating clockwise (CW), which causes the cell to tumble (*Figure 1A*).

To describe the veto model mathematically, we write the expression for the run bias (the fraction of time that a cell spends running) as a function of the CCW bias (the fraction of time that each flagellum spends rotating CCW). Assuming that the CCW bias is fixed in time, one obtains:

$$Run\,bias = \frac{Time\,ALL\,motors\,CCW}{Total\,time} \qquad (S1)$$

$$Run\,bias = (CCW\,bias)^{N_{flag}} \qquad (S2)$$

where $N_{flag}$ is the number of flagella. We can write the tumble bias (TB) as a function of the run bias, and the CW bias (CB) as a function of CCW bias:

$$TB = 1 - Run\,Bias \qquad (S3)$$

$$CB = 1 - CCW\,Bias \qquad (S4)$$

We then use these relations to solve for the tumble bias as a function of CW bias and the number of flagella:

$$TB = 1 - (1 - CB)^{N_{flag}} \qquad (S5)$$

## Simulating the effect of CheY-P fluctuations

Stochastic simulations were performed to model the effects of fluctuations in CheY-P concentration over time, using a method similar to *Sneddon, et al. (2012)*. Simulations are illustrated in *Figure 4A*. CheY-P fluctuations were generated using *Equation S6* below (see *Table 3* for parameter values), where $Y$ is the instantaneous concentration of CheY-P, $\tau$ is the characteristic timescale of fluctuations in [CheY-P], $\sigma^2$ is the variance in [CheY-P], and $\xi$ are normally distributed random numbers with unit variance. The time resolution of all simulations was one data point per millisecond.

$$Y(t + dt) = Y(t) - \frac{(Y(t) - (Y))}{\tau}dt + \sigma\sqrt{\frac{2dt}{\tau}}\xi \qquad (S6)$$

The CW bias was calculated from [CheY-P] using the Hill function relationship determined by *Cluzel et al. (2000)*. After generating the CheY-P time trace, the mean CW bias of that trace was checked, to ensure that the bias was within the same range as the data mean (0.13–0.145). The rotational state of each flagellar motor was then determined stochastically, using *Equation 1* in Sneddon et al., the CW bias, and characteristic motor switching rate ($\omega$) to set the CW/CCW transition rates ([*Sneddon et al., 2012*] see *Table 3* for parameter values).

Runs and tumbles were determined using the veto model: whenever any flagellum was rotating CW the cell tumbled, when all flagella were rotating CCW the cell ran. Finally, the simulated data were analyzed in the exact same manner as the experimental data, to extract the mean number of flagella participating in tumbles (*Figure 2C*), flagella cross-correlation (*Figure 2D*), the effective number of flagella ($N_{eff}$, *Figure 3C*) and the deviation from veto model $\eta$, (*Figure 4C*).

The values of $\sigma$ and $\tau$, which denote the amplitude and timescale of simulated CheY-P fluctuations, were determined by scanning through a range of values, and minimizing the global, reduced $\chi^2$ (*Bevington and Robinson, 2003*) from comparisons of simulations to data. The minimum reduced $\chi^2$ value indicates that all of the experimental data is best reproduced by simulations in which $\sigma = 1.0\,\mu M^2$ and $\tau = 0.2\,s^{-1}$. The global reduced $\chi^2$ was calculated by summing the individual reduced $\chi^2$ values from fits to data in *Figures 2C,D and 4C*.

## Incorporating the effect of *curly-1* flagella during runs

To test the effect of runs involving *curly-1* flagella, we augmented our simulations following *Sneddon et al. (2012)*. For each simulated flagellum, the binary CCW/CW trace was converted into a 3-state (*normal, semi-coiled, curly-1*) trace, using $\lambda^{-1} = 0.68\,s^{-1}$ as the transition rate from *semi-coiled* to *curly-1* (*Table 3*). The veto model was then applied, with the caveat that cells ran 18% of the time when there was a single *curly-1* flagellum, provided the other flagella were all CCW (*normal*) (*Figure 1—figure supplement 5*).

## Acknowledgements

We are grateful to Howard Berg, Linda Turner, Vedhavalli Nathan and Philippe Cluzel for advice and for providing reagents. We thank current and former members of the Chemla, Golding, and Rao Laboratories for providing help with experiments.

## Additional information

### Funding

| Funder | Grant reference number | Author |
| --- | --- | --- |
| National Science Foundation | PHY-082265 | Ido Golding, Yann R Chemla |
| Burroughs Wellcome Fund | | Yann R Chemla |
| Alfred P. Sloan Foundation | | Yann R Chemla |
| National Institutes of Health | R01 GM082837 | Ido Golding |
| Welch Foundation | Q-1759 | Ido Golding |
| National Science Foundation | PHY-1147498 | Ido Golding |
| National Institutes of Health | R01 GM054365 | Chris V Rao |

The funders had no role in study design, data collection and interpretation, or the decision to submit the work for publication.

### Author contributions

PJM, Conception and design, Acquisition of data, Analysis and interpretation of data, Drafting or revising the article, Contributed unpublished essential data or reagents; SK, CVR, Conception and design, Drafting or revising the article, Contributed unpublished essential data or reagents; IG, YRC, Conception and design, Analysis and interpretation of data, Drafting or revising the article

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
