## [Decision Letter]

Thank you for sending your work entitled “A simple mapping between cell swimming behavior and single-motor state in multi-flagellated *E. coli*” for consideration at *eLife*. Your article has been favorably evaluated by a Senior editor and 3 reviewers, one of whom is a member of our Board of Reviewing Editors.

The following individuals responsible for the peer review of your submission have agreed to reveal their identity: Michael Laub (Reviewing editor); Howard Berg (peer reviewer).

The Reviewing editor and the other reviewers discussed their comments before we reached this decision, and the Reviewing editor has assembled the following comments to help you prepare a revised submission.

In general, all three reviewers enjoyed reading this manuscript and the consensus was that it should ultimately be published in *eLife*; we thought it was a well-executed and important study with broad appeal. We had only a few minor issues to address in a revised manuscript:

1) Change the title to something that better captures the big picture, take-home message of the paper. The current title seems tailored for chemotaxis aficionados. Perhaps something along the lines of what was submitted as the “Impact Statement” which was “The swimming behavior of *E. coli* is surprisingly robust against natural variations in the number of flagella per cell”. Maybe just remove the word ‘surprisingly’ and it would be a title that captures the major conclusion of the paper and might appeal to a broader audience.

2) It was noted that the generalized veto model holds only up to 4 flagella per cell. It would be prudent to note this and discuss the disagreement when there are 5 or more flagella.

3) One reviewer thought it would be useful to provide a simple empirical formula for calculating *N*_*eff*_ from *N*_*flag*_ so that modellers in particular could implement the conversion as needed.

---

## [Author Response]

*1) Change the title to something that better captures the big picture, take-home message of the paper. The current title seems tailored for chemotaxis aficionados. Perhaps something along the lines of what was submitted as the “Impact Statement” which was “The swimming behavior of E. coli is surprisingly robust against natural variations in the number of flagella per cell”. Maybe just remove the word ‘surprisingly’ and it would be a title that captures the major conclusion of the paper and might appeal to a broader audience*.

We thank the reviewers for this suggestion. We have changed our title to capture the big-picture message of the paper better. The new title is “*Escherichia coli* swimming is robust against variations in flagellar number.”

*2) It was noted that the generalized veto model holds only up to 4 flagella per cell. It would be prudent to note this and discuss the disagreement when there are 5 or more flagella*.

This is a good point. In Figure 3, both strains deviate from the generalized veto models for cells with more than 4 flagella. We have added a discussion of the possible explanations for this deviation in the text.

It is possible that hydrodynamic effects must be taken in consideration in cells with many flagella. Hydrodynamic coupling in the Hu model leads to a lower *N*_*eff*_, in the direction of the deviation. Alternatively, a mechanism as described by Sneddon et al. in which cells with many flagella can run while some of its flagella rotate CW in the *curly-1* state, could lead to a similar deviation (see Figure 1—figure supplement 5). Finally, we must consider the possibility that the apparent deviation is due to systematic experimental error, since it is more difficult to determine visually the state of each flagellum on cells with many flagella.

*3) One reviewer thought it would be useful to provide a simple empirical formula for calculating N*_*eff*_
*from N*_*flag*_
*so that modellers in particular could implement the conversion as needed*.

We performed a fit to the wild-type data in Figure 3 to provide a simple empirical formula for calculating *N*_*eff*_ from *N*_*flag*_. The power law *N*_*eff*_ = 1.27 x *N*_*flag*_^0.5^ provides a reasonable match to the data and is provided in the Results section as part of the description of Figure 3. Additionally, we have added Figure 3—figure supplement 1 showing this empirical curve with the wild-type data.